# Mesothelin Gene Variants Affect Soluble Mesothelin-Related Protein Levels in the Plasma of Asbestos-Exposed Males and Mesothelioma Patients from Germany

**DOI:** 10.3390/biology11121826

**Published:** 2022-12-14

**Authors:** Hans-Peter Rihs, Swaantje Casjens, Irina Raiko, Jens Kollmeier, Martin Lehnert, Kerstin Nöfer, Kerstin May-Taube, Nina Kaiser, Dirk Taeger, Thomas Behrens, Thomas Brüning, Georg Johnen

**Affiliations:** 1Institute for Prevention and Occupational Medicine of the German Social Accident Insurance (IPA), Institute of the Ruhr University Bochum, 44789 Bochum, Germany; 2Helios Klinikum Emil von Behring GmbH, Lungenklinik Heckeshorn, Walterhöferstr. 11, 14165 Berlin, Germany

**Keywords:** asbestos, MoMar cohort, prospective study, prediagnostic mesothelioma, manifest mesothelioma, mesothelin, variant, confounder, blood test, biomarker

## Abstract

**Simple Summary:**

The continuing increase in mortality from malignant mesothelioma, often a late effect of asbestos exposure, has gained much public attention. Malignant mesothelioma is a devastating disease with limited therapeutic options. A major problem is that this cancer is usually diagnosed when the tumor is already large and has spread. An earlier diagnosis could be possible with blood tests that determine biomarkers like the protein mesothelin. The corresponding gene of mesothelin, however, can harbor genetic variants that could influence the proteins blood concentrations. We therefore studied four genetic variants in 410 asbestos-exposed males without cancer and 102 mesothelioma cases and revealed that the mesothelin concentration between the groups was significantly different (*p* < 0.0001) and that five to eight mutations of the four variants studied were associated with increased mesothelin concentrations (*p* = 0.001). These results may be a helpful tool to explain unusually high values of mesothelin protein in healthy people and provides a basis to consider the exclusion of influencing factors for an improvement of the diagnostic procedure. Finally, knowledge about confounders can be integrated into surveillance programs offered to high-risk groups of asbestos-exposed workers.

**Abstract:**

Malignant mesothelioma (MM) is a severe disease mostly caused by asbestos exposure. Today, one of the best available biomarkers is the soluble mesothelin-related protein (SMRP), also known as mesothelin. Recent studies have shown that mesothelin levels are influenced by individual genetic variability. This study aimed to investigate the influence of three mesothelin (*MSLN*) gene variants (SNPs) in the 5′-untranslated promoter region (5′-UTR), *MSLN rs2235503 C > A*, *rs3764246 A > G*, *rs3764247 A > C*, and one (*rs1057147 G > A*) in the 3′-untranslated region (3′-UTR) of the *MSLN* gene on plasma concentrations of mesothelin in 410 asbestos-exposed males without cancer and 43 males with prediagnostic MM (i.e., with MM diagnosed later on) from the prospective MoMar study, as well as 59 males with manifest MM from Germany. The mesothelin concentration differed significantly between the different groups (*p* < 0.0001), but not between the prediagnostic and manifest MM groups (*p* = 0.502). Five to eight mutations of the four SNP variants studied were associated with increased mesothelin concentrations (*p* = 0.001). The highest mesothelin concentrations were observed for homozygous variants of the three promotor SNPs in the 5′-UTR (*p* < 0.001), and the highest odds ratio for an elevated mesothelin concentration was observed for *MSLN rs2235503 C > A*. The four studied SNPs had a clear influence on the mesothelin concentration in plasma. Hence, the analysis of these SNPs may help to elucidate the diagnostic background of patients displaying increased mesothelin levels and might help to reduce false-positive results when using mesothelin for MM screening in high-risk groups.

## 1. Introduction

Malignant mesothelioma (MM) is a fatal disease caused by asbestos exposure as a major risk factor [1,2]. Although there is an urgent need for non-invasive detection methods, such markers are scarce [3]. Today, one of the best available serum markers is the soluble mesothelin-related protein (SMRP), also known as mesothelin [1,4,5]. Mesothelin, in combination with calretinin, is the only marker so far that has shown potential for the early detection of MM when tested in serial blood samples from prospective asbestos cohorts [6,7].

Several efforts have been made to conduct research on other proteins of the mesothelin family and to study their potential as MM biomarkers. One of these is the megakaryocyte potentiating factor (MPF), also known as N-ERC/mesothelin [8,9]. Mesothelin and MPF are products of the mesothelin gene *MSLN.* The *MSLN* gene encodes a 71-kDa precursor protein, which is physiologically cleaved by furin-like proteases into two fragments. The C-terminal 40-kDa fragment remains membrane-bound and is classified as mature mesothelin. The 31-kDa soluble N-terminal fragment, called MPF or N-ERC/mesothelin, is secreted into the blood [8,10]. Recently, we showed that a recombinant polypeptide of MPF could be a cost-effective and minimally invasive contribution to support a diagnosis of MM, especially in regions with limited medical care [11,12]. Despite these efforts, there are several confounding factors that influence diagnostic outcomes. Knowledge of the influencing factors could help us understand the limitations of biomarkers and might also be used to improve their specificity. A wide variety of factors are known to influence mesothelin. For example, Scherpereel et al. [13] showed that tumor histology has an influence on the impact of diagnostics. Furthermore, Casjens et al. [14,15] reported that renal dysfunction, bronchitis, age, hypertension, and elevated inflammation values may also affect the accuracy of mesothelin as a diagnostic marker. Finally, a third factor that has to be taken in account is the genetic background. The NHLBI Exome Sequencing Project (ESP) currently displays 217 variations in the *MSLN* gene for European/American populations. Several studies from Italy [16,17,18,19] and one each from Slovenia [20] and China [21] showed strong evidence that certain single-nucleotide polymorphisms (SNPs) in the *MSLN* promoter region and also the SNP *MSLN rs1057147* have a modulating influence on the diagnostic impact of mesothelin levels. In this target region, multiple other SNPs exist, but a general problem in the promoter region of the *MSLN* gene is the presence of SNPs that are in a linkage disequilibrium. Examples are rs2235503 with rs12597489 and rs3764246 with both rs2235504 and, to a lesser extent, rs2235505 [16].

Therefore, the aim of this study was to investigate the influence of the four most common *MSLN*-SNPs (three in the 5′-UTR promoter region and one in the 3′-UTR of the *MSLN* gene), without close linkage disequilibrium with each other, on the mesothelin levels measured in plasma obtained from male participants of the prospective MoMar study, extending the findings from Italy and Slovenia to a German asbestos cohort. Another aim was the evaluation of *MSLN*-SNPs as influencing factors in the context of future MM screening programs.

## 2. Methods

### 2.1. Study Population

The study population comprised 410 asbestos-exposed but cancer-free male controls with benign asbestos-related diseases recognized as occupational diseases in Germany. This included diagnoses of pneumoconiosis, pleural plaques, pleural effusions or pleuritis, asbestosis, pleural thickening, and/or pleural fibrosis [22]. In addition, 102 men with MM, of whom 43 were prediagnostic and 59 manifest MM cases, were included. Control subjects were part of the Molecular Marker (MoMar) prospective cohort and were randomly selected according to age and smoking status out of 2769 study participants. Study subjects were recruited at medical offices participating in the MoMar study and had a follow-up of up to ten years [6,22,23]. The 43 prediagnostic MM cases were also part of the MoMar study, and all samples were drawn before diagnosis (median 8.5 months, interquartile range (IQR) 4.5 to 13 months). In contrast, the 59 manifest MM cases were recruited at the “Lungenklinik Heckeshorn, Helios Klinikum Emil von Behring” in Berlin. Samples of manifest MM cases were drawn a median of 0.5 months after diagnosis (IQR 0.2 to 0.9 months). Information on smoking habits as well as current and chronic diseases (hypertension, diabetes mellitus, rheumatoid arthritis, liver disease, intestinal disease, renal insufficiency) was derived from questionnaires for all subjects. All subjects provided written informed consent. The study was approved by the ethics committee of the Ruhr University Bochum (reference number 3217-08).

### 2.2. Genotyping of Four MSLN Single-Nucleotide Polymorphisms (SNPs) and Mesothelin Measurements in Plasma

EDTA blood samples were taken on the day of the medical examination and separated into plasma and cellular fraction by centrifugation (10 min at 2000× *g*) within 30 min. Both components were immediately frozen and then sent to the IPA in Bochum, where they were aliquoted and stored at −80°C until use. Genomic DNA was purified from the cellular fraction in a QIA cube or manually using the QIA amp blood kit in accordance with the protocols of the supplier (Qiagen, Hilden, Germany). Real-time PCR analyses for three *MSLN* gene SNPs (*rs2235503 C > A*, *rs3764246 A > G*, *rs3764247 A > C*) in the promoter region placed closest to the transcription start site and one polymorphism (*rs1057147 G > A*) in the 3′-UTR placed behind the *MSLN* gene were analyzed on a LightCycler 2.0 instrument (Roche, Penzberg, Germany). Four different on-demand LightSNiP assays and the corresponding protocols from TIB MOLBIOL GmbH (Berlin, Germany) were used to analyze the status of these SNPs. Mesothelin in plasma samples was measured using an ELISA kit MESOMARK™ (Fujirebio Diagnostics, Inc., Malvern, PA, USA) in accordance with the supplier’s protocol. Mesothelin concentrations equal or above the chosen cut-off of 2.9 nM were considered as positive test results [6]. Hence, mesothelin ≥ 2.9 nM displayed a false-positive result in asbestos-exposed cancer-free controls. This stringent cut-off was adopted from the early detection setting of the MoMar cohort study, which required a high specificity for the screening of the rare tumor entity MM [6].

### 2.3. Statistical Analyses

Box plots with median and interquartile ranges (IQRs) were used to depict the distribution of mesothelin concentrations. Whiskers represent the minimum and maximum. Mann–Whitney U tests or, in cases involving more than two groups, Kruskal–Wallis tests were applied to examine group differences. Fisher’s exact test was used to examine dependencies between two categorical variables. *p* values < 0.05 were considered statistically significant. Odds ratios (ORs) with 95% confidence intervals (CIs) were estimated to assess the risk of mesothelin concentrations being equal to or above the cut-off of 2.9 nM using multiple logistic regression analyses. As potential predictors, we examined group status (controls, prediagnostic MM, manifest MM); the number of mutations; haplotype; and SNP status. Due to the small participant numbers, in some cases only models with two SNP categories (common and non-common genotype) could be calculated. All models were adjusted for age. The Akaike Information Criterion (AIC) was used as a selection criterion for the statistical models. The smaller the AIC value, the better the models’ goodness of fit. All statistical analyses were performed using SAS software, version 9.4 (SAS Institute Inc., Cary, NC, USA). Graphs were prepared with GraphPad Prism version 7.04 (GraphPad Software, La Jolla, CA, USA).

## 3. Results

### 3.1. Characterization of the Study Population

The main characteristics of the study population comprising 512 men are summarized in Table 1, including 410 asbestos-exposed but cancer-free controls and 43 prediagnostic and 59 manifest MM cases. The study groups did not differ with respect to age (overall median: 73 years) or smoking status. The most frequently self-reported diseases were hypertension, with a total of 315 cases (62%), and diabetes mellitus, with a total of 77 cases (15%). Controls had lower mesothelin concentrations than MM cases (*p* < 0.0001). The mesothelin concentrations of the manifest MM cases were slightly higher than those of the prediagnostic MM cases (1.61 nM vs. 1.34 nM), but this difference was not statistically significant (*p* = 0.502).

### 3.2. Distribution of False-Positive and False-Negative Mesothelin Values by Single-Nucleotide Polymorphisms (SNPs) and Haplotype

Table 2 shows the distribution of false- and true-positive mesothelin tests stratified by study group and as a function of SNPs. It demonstrates again that manifest MM cases presented a true-positive test result more frequently than prediagnostic MM cases (23.7% vs. 18.6%), but without statistical significance (chi^2^ test: *p* = 0.534, data not shown). False-positive test results occurred especially frequently in homozygous variants, which was also reflected by the statistical models (Table 3). Mutations in the SNPs increased the OR for a false-positive marker result markedly, and the highest OR was observed when a double mutation of the analyzed SNP was found. For example, when examining the *MSLN-SNP rs2235503* C > A, the OR for a false-positive marker was 20.7-fold higher when the double mutation *AA* was present compared to the common genotype. The OR for a positive test result also increased in the prediagnostic cases in the presence of a mutation, but not in manifest cases (Table 3). In prediagnostic MM cases, for example, the *MSLN-SNP* rs3764246A > G showed a 33.3-fold OR, and the *MSLN-SNP rs3764247A > C* showed a 22.7-fold OR for a false-positive marker for the corresponding double mutations *GG* and *CC*, respectively, when compared with the common genotype in this subgroup.

### 3.3. Distribution of the Four Single-Nucleotide Polymorphism Mutations by Study Group

Figure 1 graphically depicts that mutations were significantly more frequent in (false-positive) controls (*p* = 0.004) and prediagnostic MM cases (*p* = 0.006) compared with manifest MM cases and were particularly more frequent when the mesothelin level was ≥2.9 nM.

### 3.4. Distribution of the Mesothelin Concentration between 5′-UTR Haplotypes and Stratified by Study Groups

Figure 2 shows the distribution of mesothelin concentration between 5′-UTR haplotypes and stratified by study groups. The median concentration was lowest with no mutation (haplotype AAC) in all study groups. In the presence of two (haplotypes CAA and CGC) or three (haplotype CGA) mutations, the median mesothelin concentration increased and was more often above the cut-off of 2.9 nM. In the presence of haplotype CGA with three mutations, the cut-off was exceeded most frequently. Of the 33 cases in which the mesothelin cut-off was exceeded, 15 occurred in samples with the haplotype CGA (45%). This was particularly true for asbestos-exposed controls (73%) and for prediagnostic MM cases (75%). In contrast, ten of the fourteen manifest MM cases that exceeded the cut-off had no mutation (71%), and just one sample with the haplotype CGA exceeded the cut-off. The corresponding Fisher’s exact tests revealed that the haplotypes among the participants with positive mesothelin test results were not equally distributed across the study groups (*p* = 0.0015).

### 3.5. Logistic Regression Models in Asbestos-Exposed Controls and MM Groups Adjusted by Age

Appendix A shows the impact of the study groups and SNPs on elevated mesothelin concentrations assessed with multiple logistic regression models. The OR for a positive test result was 10-fold higher for MM than for the controls (OR = 10.44, 95% CI 4.84–22.5, data not shown), with slightly higher estimates in manifest cases (OR = 12.70, 95% CI 5.36–30.1) than in prediagnostic cases (OR = 7.98, 95% CI 3.00–21.2). In addition to group status, the consideration of the haplotype or the number of mutations resulted in a better model fit, as indicated by a smaller AIC. Adding the *MSLN-SNP rs1057147* from the 3′-UTR to the three SNPs from the 5′-UTR did not improve the model fit.

## 4. Discussion

MM is a disease with limited treatment options and a low survival rate, not least because it is usually detected at a late stage. Biomarkers might improve outcomes if they can facilitate an earlier diagnosis at a less advanced stage with better treatment options. However, in a disease like MM, it is important to avoid false-positive biomarker results in order to limit unnecessary psychological burdens or potentially harmful procedures during diagnostic follow-up. Additionally, MM is a rare disease, generally resulting in low positive predictive values for biomarkers. Knowledge about its possible influencing factors—such as a reduced glomerular filtration rate—could help to improve the performance of biomarkers, e.g., by excluding persons with kidney failure when screening for MM in high-risk groups. A specific genetic background could be another reason for misleading elevated biomarker levels. We therefore examined different *MSLN-*SNPs in MM cases and controls in the context of their mesothelin protein levels in corresponding plasma samples from a prospective asbestos cohort.

In asbestos-exposed controls, the number of mutations was clearly increased in the subgroup of men with mesothelin concentrations ≥ 2.9 nM (Figure 1, Table 2 and Appendix A), i.e., persons who were defined as false positives in the MoMar cohort study [6]. Additionally, the prediagnostic MM cases with true-positive results showed an analogous picture. In contrast, we could not observe an association between mesothelin concentration and the number of mutations in the manifest MM cases. A previous study from Italy also showed an association between serum mesothelin concentration and variant alleles in controls but not in MM cases [18]. However, it was not reported whether prediagnostic or manifest MM cases were examined. Without differentiating MM subgroups in the present study, the number of mutations also did not differ between MM cases with high or low mesothelin concentrations (*p* = 0.153, data not shown). As we considered two MM subgroups and applied a relatively stringent cut-off of 2.9 nM, which was defined for the early detection setting in the MoMar cohort study, it was difficult to compare our observations with the results of earlier Italian studies [16,17,18,19]. This relatively stringent cut-off was chosen to pre-emptively limit the influence of possible confounders and thus reach a high specificity of 99% for mesothelin in the MoMar cohort comprising patients with benign asbestos-related diseases [6]. The high specificity resulted in a lower sensitivity for detecting MM. Our strategy was to compensate for the low sensitivity by combining mesothelin with one or more other highly specific markers, e.g., calretinin [6]. An individualized, higher cut-off for persons with variant alleles was suggested by Cristaudo et al. [16]. Regarding *MSLN*-SNP *rs1057147,* the SNP from the 3′-UTR, Goricar et al. [20] reported that heterozygotes and carriers of two polymorphic alleles had significantly higher SMRP levels among 628 subjects without MM but not in the 154 studied MM patients. The 399 true-negative controls in our study displayed a comparable distribution. The *GA* frequency in the German controls was 33.1% vs. 34.6% in the 628 Slovenian subjects without MM [20]. The *AA* frequency also revealed comparable percentages (6.5% vs. 6.0%).

In the presence of the haplotype CGA with mutations at the three nucleotide positions −724, −621, −171 in front of the translation start site in the 5′-UTR MSLN promoter region, the mesothelin cut-off in our study was most frequently exceeded in the asbestos-exposed controls as well as in the prediagnostic MM cases, but not in the manifest cases. This observation was in good agreement with the results of in vitro studies performed by De Santi et al. [18] and Silvestri et al. [19]. Both suggested that the CGAG haplotype that included a fourth *MSLN*-SNP rs2235504 in nucleotide position −109 could enhance the activity of the MSLN promoter. In our study we did not consider this SNP due to the strong linkage disequilibrium with *MSLN*-SNP rs3764246 in nucleotide position −621.

The identification of influencing factors could help to explain and ultimately reduce false-positive results in biomarker screening. In practical applications, all samples with positive results would be checked for possible confounders, e.g., renal failure by determining the glomerular filtration rate [24] or variant alleles by genotyping. Whether the regular analysis of gene variants would be a practical and cost-effective method in clinical practice remains to be determined. One has to keep in mind, however, that prediagnostic MM cases—whose detection would be the goal of a biomarker screening—also showed the accumulation of *MSLN* mutations. In their early stages, MM tumors can sometimes be too small to detect and confirm by imaging. These patients would therefore be indistinguishable from cancer-free persons with false-positive marker results. A possible solution would be to establish a baseline with a timeline of mesothelin concentrations for each individual in a screening program to distinguish between real MM cases (increase in mesothelin with time) and persons with constitutively elevated levels (mesothelin concentrations relatively steady). This approach is comparable to CT screening, where conspicuous nodules are re-examined at a later time to detect malignant growth [25].

A limitation of this study was the relatively small number of MM cases, particularly prediagnostic cases. The phenomena seen in this subgroup might disappear in a larger collective. However, MM is a rare tumor and mostly detected in its later stages. Therefore, the only sources of prediagnostic samples from MM patients are large longitudinal cohort studies with serial sampling. Unfortunately, studies of this kind are scarce. A strength of this study was the utilization of samples from such a cohort, which had the advantage that cases and controls were derived from the same target population, thereby limiting bias.

## 5. Conclusions

This study explored four *MSLN*-*SNP* variants as potential predictors associated with false-positive results for a blood-based MM marker in plasma and confirmed the results of previous studies obtained with different populations. The early detection of MM requires high-specificity biomarkers. The determination of the SNPs studied here may be a helpful tool to explain unusually high soluble mesothelin-related protein levels and provide a basis for considering the exclusion of influencing factors for the further improvement of diagnostic procedures. Knowledge of the confounding factors could be integrated into surveillance programs offered to high-risk groups of asbestos-exposed workers. A pilot program involving the use of mesothelin and calretinin for the early detection of MM is currently planned for implementation in Germany.

## Figures and Tables

**Figure 1 biology-11-01826-f001:**
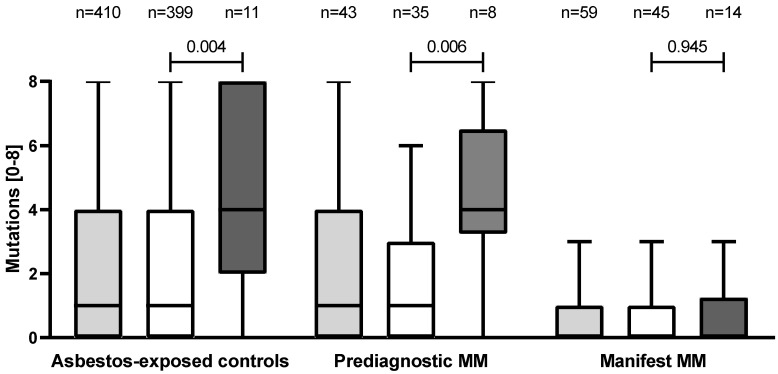
Distribution of the number of single-nucleotide polymorphism mutations by study group. *p* values of Mann–Whitney U tests displayed mutation differences between groups with low (<2.9 nM, white boxes) and high (≥2.9 nM, dark grey boxes) concentrations of mesothelin. Light grey boxes depict the number of single-nucleotide polymorphism mutations in each study group without distinguishing mesothelin concentrations.

**Figure 2 biology-11-01826-f002:**
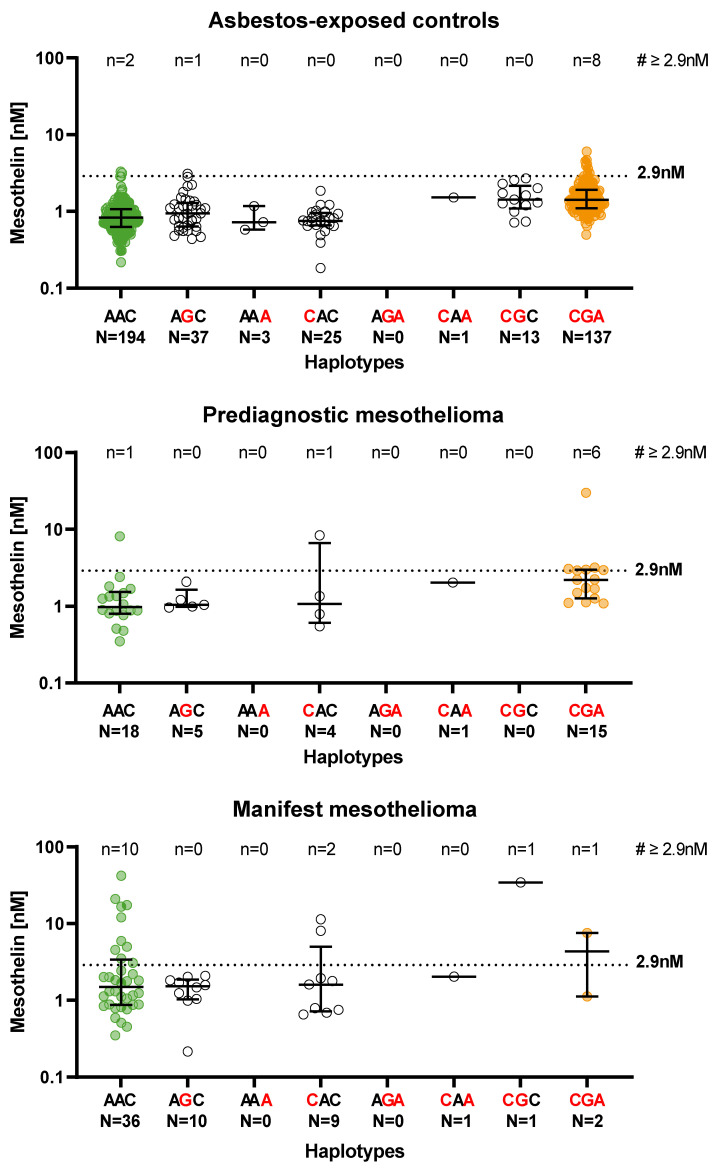
Association between haplotypes and mesothelin concentrations in 512 males stratified by study group. Samples without mutation (haplotype AAC) are marked in green, samples with one or two mutations are marked in white, and samples with three mutations (haplotype CGA) are marked in orange.

**Table 1 biology-11-01826-t001:** Characteristics of the study population comrpising 410 asbestos-exposed controls and 102 mesothelioma (MM) patients.

		Asbestos-Exposed Controls	Prediagnostic MM	Manifest MM	
		n (%)	n (%)	n (%)	*p* Value ^c^
N		410	43	59	
Mesothelin (nM)	Median (IQR ^b^)	1.02 (0.74–1.41)	1.34 (0.90–2.21)	1.61 (0.88–2.43)	<0.001
Age (years)	Median (IQR ^b^)	73 (69–77)	74 (71–77)	72 (66–75)	0.206
Smoking status	Never	129 (31.5)	12 (27.9)	22 (37.3)	0.791
	Former	233 (56.8)	27 (62.8)	32 (54.2)	
	Current	44 (10.7)	4 (9.3)	4 (6.8)	
	Missing	4 (1.0)	0 (0)	1 (1.7)	
Hypertension ^a^	Yes	262 (63.9)	26 (60.5)	27 (45.8)	0.082
Diabetes mellitus ^a^	Yes	61 (14.9)	7 (16.3)	9 (15.3)	0.244
Rheumatoid arthritis ^a^	Yes	17 (4.2)	2 (4.7)	1 (1.7)	0.167
Liver disease ^a^	Yes	8 (2.0)	2 (4.7)	2 (3.4)	0.076
Intestinal disease ^a^	Yes	10 (2.4)	3 (7.0)	1 (1.7)	0.055
Renal insufficiency ^a^	Yes	6 (1.5)	1 (2.3)	2 (3.4)	0.092

^a^ Disease information could not be determined for all 512 men. ^b^ Interquartile range. ^c^ Comparison of all controls and two mesothelioma groups tested with Kruskal–Wallis tests or Fisher tests.

**Table 2 biology-11-01826-t002:** Distribution of false-positive and false-negative mesothelin values by single-nucleotide polymorphisms (SNPs) and haplotype.

	Asbestos-Exposed Controls (N = 410)						Prediagnostic Mesothelioma (N = 43)						Manifest Mesothelioma (N = 59)					
	Mesothelin < 2.9 nM (True-Negative)			Mesothelin ≥ 2.9 nM (False-Positive)			Mesothelin < 2.9 nM (False-Negative)			Mesothelin ≥ 2.9 nM (True-Positive)			Mesothelin < 2.9 nM (False-Negative)			Mesothelin ≥ 2.9 nM (True-Positive)		
	N = 399 (97.3%)			N = 11 (2.7%)			N = 35 (81.4%)			N = 8 (18.6%)			N = 45 (76.3%)			N = 14 (23.7%)		
	N	%	95% CI	N	%	95% CI	N	%	95% CI	N	%	95% CI	N	%	95% CI	N	%	95% CI
SNPs from 5′-UTR																		
MSLN rs3764247 A > C																		
AA	231	57.9	53.1–62.7	3	27.3	1.0–53.6	22	62.9	46.8–78.9	1	12.5	0–35.4	36	80.0	68.3–91.7	10	71.4	47.8–95.1
A**C**	141	35.3	30.6–40.0	4	36.4	7.9–64.8	10	28.6	13.6–43.5	4	50.0	15.4–84.6	8	17.8	6.6–28.9	3	21.4	0–42.9
CC	27	6.8	4.3–9.2	4	36.4	7.9–64.8	3	8.6	0–17.8	3	37.5	4.0–71.0	1	2.2	0–6.5	1	7.1	0–20.6
MSLN rs3764246 A > G																		
AA	221	55.4	50.5–60.3	2	18.2	0–41.0	21	60.0	43.8–76.2	2	25.0	0–55.0	34	75.6	63–88.1	12	85.7	67.4–100
A**G**	141	35.3	30.6–40.0	4	36.4	7.9–64.8	13	37.1	21.1–53.2	4	50.0	15.4–84.6	11	24.4	11.9–37.0	2	14.3	0–32.6
GG	37	9.3	6.4–12.1	5	45.5	16.0–74.9	1	2.9	0–8.4	2	25.0	0–55.0	0			0		
MSLN rs2235503 C > A																		
CC	266	66.7	62.0–71.3	3	27.3	1.0–53.6	25	71.4	56.5–86.4	2	25.0	0–55.0	43	95.6	89.5–100	13	92.9	79.4–100
C**A**	117	29.3	24.9–33.8	4	36.4	7.9–64.8	10	28.6	13.6–43.5	4	50.0	15.4–84.6	2	4.4	0–10.5	1	7.1	0–20.6
AA	16	4.0	2.1–5.9	4	36.4	7.9–64.8	0			2	25.0	0–55.0	0			0		
Haplotype of the three SNPs from 5′-UTR																		
No mutation (AAC)	192	48.1	43.2–53.0	2	18.2	0–41.0	17	48.6	32–65.1	1	12.5	0–35.4	26	57.8	43.3–72.2	10	71.4	47.8–95.1
1–2 mutations *	78	19.5	15.7–23.4	1	9.1	0–26.1	9	25.7	11.2–40.2	1	12.5	0–35.4	18	40.0	25.7–54.3	3	21.4	0–42.9
3 mutations (**CGA**)	129	32.3	27.7–36.9	8	72.7	46.4–99.0	9	25.7	11.2–40.2	6	75.0	45–100	1	2.2	0–6.5	1	7.1	0–20.6
MSLN rs1057147 G > A from 3′-UTR																		
GG	243	60.9	56.1–65.7	3	27.3	1.0–53.6	22	62.9	46.8–78.9	1	12.5	0–35.4	38	84.4	73.9–95.0	12	85.7	67.4–100
G**A**	132	33.1	28.5–37.7	5	45.5	16.0–74.9	10	28.6	13.6–43.5	5	62.5	29–96	5	11.1	1.9–20.3	2	14.3	0–32.6
AA	24	6.0	3.7–8.3	3	27.3	1.0–53.6	3	8.6	0–17.8	2	25.0	0–55.0	2	4.4	0–10.5	0		

* A**G**C, AA**A**, **C**AC, A**GA**, **C**A**A**, **CG**C (mutations in bold).

**Table 3 biology-11-01826-t003:** Odds ratios (ORs) and 95% confidence intervals (CIs) for predictors of increased mesothelin concentrations in men assessed with logistic regression models adjusted for age.

Model	Effect	Asbestos-Exposed Controls (N = 410)				Prediagnostic MM (N = 43)				Manifest MM (N = 59)			
		OR	95% CI		*p* Value	OR	95% CI		*p* Value	OR	95% CI		*p* Value
Age (years)		1.10	1.00	1.23	0.060	1.06	0.90	1.24	0.515	1.03	0.97	1.10	0.312
Haplotype (3 SNPs of the 5′-UTR)	3 mutations	**6.00**	**1.25**	**28.90**	**0.025**	**11.03**	**1.13**	**107.5**	**0.039**	1.69	0.09	32.20	0.728
Ref ^b^: no mutation	1–2 mutations	1.29	0.11	14.54	0.835	1.91	0.11	34.43	0.660	0.35	0.08	1.53	0.164
MSLN rs3764247 A > C (ref ^b^: AA)	AC	2.06	0.45	9.40	0.350	8.32	0.81	85.23	0.074	1.22	0.26	5.58	0.802
	CC	**10.83**	**2.26**	**51.99**	**0.003**	**22.74**	**1.72**	**300.1**	**0.018**	2.94	0.16	52.69	0.464
MSLN rs3764247 A > C (ref ^b^: common genotype)	Non-common genotype	3.45	0.90	13.30	0.072	**11.56**	**1.27**	**105.2**	**0.030**	1.42	0.35	5.73	0.625
MSLN rs3764246 A > G (ref ^b^: AA)	AG	3.47	0.62	19.32	0.156	2.60	0.39	17.37	0.323	0.41	0.08	2.22	0.302
	GG	**15.59**	**2.87**	**84.66**	**0.001**	**33.29**	**1.57**	**703.8**	**0.024**	-^a^			
MSLN rs3764246 A > G (ref ^b^: common genotype)	Non-common genotype	**6.11**	**1.29**	**28.85**	**0.022**	4.31	0.75	24.83	0.102	0.41	0.08	2.22	0.302
MSLN rs2235503 C > A (ref ^b^: CC)	CA	3.07	0.67	13.98	0.148	-^a^				1.54	0.12	19.08	0.735
	AA	**20.68**	**4.14**	**103.4**	**<0.001**	-^a^				-^a^			
MSLN rs2235503 C > A (ref ^b^: common genotype)	Non-common genotype	**5.31**	**1.38**	**20.48**	**0.015**	**7.25**	**1.22**	**42.99**	**0.029**	1.54	0.12	19.08	0.735
MSLN rs1057147 G > A (ref ^b^: GG)	GA	3.10	0.73	13.23	0.127	**10.69**	**1.09**	**104.6**	**0.042**	-^a^			
	AA	**8.87**	**1.66**	**47.50**	**0.011**	**15.40**	**1.03**	**229.6**	**0.047**	-^a^			
MSLN rs1057147 G > A (ref ^b^: common genotype)	Non-common genotype	**4.08**	**1.06**	**15.70**	**0.041**	**11.74**	**1.29**	**106.9**	**0.029**	1.02	0.18	5.76	0.983

*MM—*malignant mesothelioma. ^a^ Modeling not possible due to small numbers; ^b^ reference. Statistically significant results are shown in bold.

## Data Availability

The data presented in this study are available on request from the corresponding author.

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
