# Peer review of "Mesothelin Gene Variants Affect Soluble Mesothelin-Related Protein Levels in the Plasma of Asbestos-Exposed Males and Mesothelioma Patients from Germany"

_biology, 2022, doi:10.3390/biology11121826_

Round 1

Reviewer 1 Report

In malignant mesothelioma screening there is a urgent need to reduce false positive results when using mesothelin. The authors want to investigate the impact of four SNPs in the MSLN gene on plasma concentration of soluble mesothelin related protein SMRP. Specific minor comments are outlined below.

1) Table 1: please add p values for comparison between groups

2) Table 2 shows results in a poorly readable way. Please improve

Author Response

Comments and Suggestions for Authors from Reviewer 1:

In malignant mesothelioma screening there is a urgent need to reduce false positive results when using mesothelin. The authors want to investigate the impact of four SNPs in the MSLN gene on plasma concentration of soluble mesothelin related protein SMRP. Specific minor comments are outlined below.

1) Table 1: please add p values for comparison between groups

2) Table 2 shows results in a poorly readable way. Please improve

Answer:

Thanks for pointing out the issues with the tables.

(1) We have added the missing p-values for comparison between groups

(2) Indeed, the readability of Table 2 has been compromised (originally, it was designed for a page in landscape format). We improved the readability of Table 2 within the limits of the available space on a page in portrait format.

Reviewer 2 Report

MM is a severe disease caused by asbestos exposure.  One of the best available biomarkers is mesothelin. The mesothelin levels are influenced by individual genetic variability. 

The authors investigated the influence of three mesothelin gene variants (SNPs) in the 5‘-UTR and one in the 3‘-UTR on plasma concentrations of mesothelin. 

They showed that the mesothelin concentration between the different groups was significantly different.  They resulted that the highest mesothelin concentrations were observed for homozygous variants of the three promotor SNPs in the 5‘-UTR.  They concluded that analyses of these SNPs may help to elucidate the diagnostic background of patients and might help to reduce false-positive results in high-risk groups.

Author Response

Comments and Suggestions for Authors from Reviewer 2:

MM is a severe disease caused by asbestos exposure. One of the best available biomarkers is mesothelin. The mesothelin levels are influenced by individual genetic variability. The authors investigated the influence of three mesothelin gene variants (SNPs) in the 5‘-UTR and one in the 3‘-UTR on plasma concentrations of mesothelin.

They showed that the mesothelin concentration between the different groups was significantly different. They resulted that the highest mesothelin concentrations were observed for homozygous variants of the three promotor SNPs in the 5‘-UTR. They concluded that analyses of these SNPs may help to elucidate the diagnostic background of patients and might help to reduce false-positive results in high-risk groups.

Answer:

Thank you for your review. We performed an additional spell check to improve the manuscript.

Reviewer 3 Report

This is a well written manuscript describing the influence of certain SNPs on plasma levels of mesothelin, which is a promising screening marker for malignant mesothelioma in high risk individuals. SNPs in the mesothelin gene may confound mesothelin-based serologic screening as these SNPs may affect serum protein levels which impacts a general robust cut-off for predicting malignant mesothelioma. The data is presented in a clear manner (although spacing of table 2 needs improved alignment, at least from the PDF that was rendered for review) and the data are discussed adequately. 

There are a few minor things I encountered which the authors may consider worth modifying.

  • I had some difficulties with the term prediagnostic as mentioned in the abstract. Although this is very well described in the methods (2.1.) the authors may consider  to add something like „i.e. with MM diagnosed in the later time course“ to make this already clear in the abstract.
  • Similarly, it was not before the discussion part of the manuscript before to understand the rationale for a 2.9nM cut-off of mesothelin levels (basically, this was adopted from the prior MoMar cohort study) and it may be worth considering to make this already more clear in the methods (2.2). This cut-off seems reasonable in the context of minimal false-positive results and the provided data demonstrate clearly that vast majority of false-positive cases in the control group are caused by SNPs (9 out of 11). However, the used cut-off unfortunately has very low sensitivity for mesothelioma detection (80%> false negative in the pre-diagnostic group and > 75% false-negative in the manifest MM group), and this lacks discussion. 
  • The authors may consider to add few additional information about SNPs in mesothelioma in the introduction. They chose to analyze the four most common ones. You can estimate the expected frequency of these four SNPs from the control group, but how many other SNPs are in the mesothelin gene? How common are other SNPs? Could they potentially also influence mesothelin levels?

Author Response

Comments and Suggestions for Authors from Reviewer 3:

This is a well written manuscript describing the influence of certain SNPs on plasma levels of mesothelin, which is a promising screening marker for malignant mesothelioma in high risk individuals. SNPs in the mesothelin gene may confound mesothelin-based serologic screening as these SNPs may affect serum protein levels which impacts a general robust cut-off for predicting malignant mesothelioma. The data is presented in a clear manner (although spacing of table 2 needs improved alignment, at least from the PDF that was rendered for review) and the data are discussed adequately.

(1) There are a few minor things I encountered which the authors may consider worth modifying. I had some difficulties with the term prediagnostic as mentioned in the abstract. Although this is very well described in the methods (2.1.) the authors may consider to add something like „i.e. with MM diagnosed in the later time course“ to make this already clear in the abstract.

(2) Similarly, it was not before the discussion part of the manuscript before to understand the rationale for a 2.9nM cut-off of mesothelin levels (basically, this was adopted from the prior MoMar cohort study) and it may be worth considering to make this already more clear in the methods (2.2). This cut-off seems reasonable in the context of minimal false-positive results and the provided data demonstrate clearly that vast majority of false-positive cases in the control group are caused by SNPs (9 out of 11). However, the used cut-off unfortunately has very low sensitivity for mesothelioma detection (80%> false negative in the pre-diagnostic group and >75% false-negative in the manifest MM group), and this lacks discussion.

(3) The authors may consider to add few additional information about SNPs in mesothelioma in the introduction. They chose to analyze the four most common ones. You can estimate the expected frequency of these four SNPs from the control group, but how many other SNPs are in the mesothelin gene? How common are other SNPs? Could they potentially also influence mesothelin levels.

Answers:

Thanks for your valuable suggestions. Besides improving the readability of Table 2 we performed the following changes:

(1) We added the following information in the abstract:
… with prediagnostic MM (i.e., MM diagnosed later in the time course) from the prospective MoMar study …

(2) In the Methods (chapter 2.2) we added the sentence:
This stringent cut-off was adopted from the early detection setting of the MoMar cohort study, which required a high specificity for the screening of the rare tumour entity MM [6].
And in the Discussion we added:
Consequently, the high specificity resulted in a lower sensitivity to detect MM. Our strategy was to compensate the low sensitivity by combining mesothelin with one or more other highly specific markers, e.g., calretinin [6].

(3) We have added now additional information concerning SNPs in the MSLN gene in the introduction. The NHLBI Exome Sequencing Project (ESP) displays actually 217 variations in the MSLN gene for European American populations. The frequency information is not available for the SNPs in ESP but in the case of our chosen three SNPs in the promoter region the minor allele frequency (MAF) is between 0.19 and 0.27 and so in the same order of magnitude as in the De Santi Paper (between 0.15 and 0.24). The potential influence on mesothelin levels from other SNPs is of course possible but is only valid when shown in appropriate experiments (in vitro or in vivo).